# Evaluating LLMs on Chinese Idiom Translation

**Cai Yang[1], Yao Dou[2], David Heineman[2], Xiaofeng Wu[2], Wei Xu[2]**

[1]Independent Contributor   [2]Georgia Institute of Technology

caiyang.cy@outlook.com, {douy,david.heineman,xwu414}@gatech.edu, wei.xu@cc.gatech.edu

## Abstract

Idioms, whose figurative meanings usually differ from their literal interpretations, are common in everyday language, especially in Chinese, where they often contain historical references and follow specific structural patterns. Despite recent progress in machine translation with large language models, little is known about Chinese idiom translation. In this work, we introduce IDIOMEVAL, a framework with a comprehensive error taxonomy for Chinese idiom translation. We annotate 900 translation pairs from nine modern systems, including GPT-4o and Google Translate, across four domains: web, news, Wikipedia, and social media. We find these systems fail at idiom translation, producing incorrect, literal, partial, or even missing translations. The best-performing system, GPT-4, makes errors in 28% of cases. We also find that existing evaluation metrics measure idiom quality poorly with Pearson correlation below 0.48 with human ratings. We thus develop improved models that achieve $F_1$ scores of 0.68 for detecting idiom translation errors.

## 1 Introduction

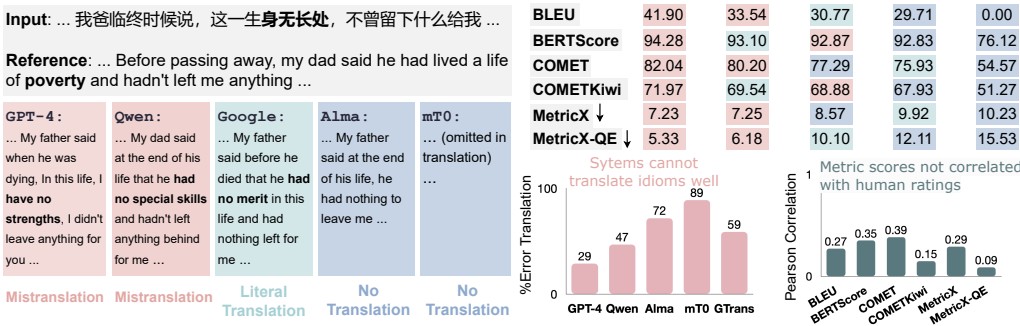

Figure 1: Top left: example of Chinese input and translations from five systems. Top right: percentage of good translations in each system. Bottom left: evaluation metric scores of system translations. Bottom right: Pearson's *r* between human rating and metric scores.

Idioms are phrases whose meanings typically differ from the literal meanings of their individual components. They are commonly used in everyday language, making their understanding crucial for tasks such as neural machine translation (Baziotis et al., 2023) and writing assistance (Tan & Jiang, 2021; Dankers et al., 2022a). In particular, idioms have posed a challenge to translation as their meaning is usually non-compositional and cannot be inferred from their literal parts. For instance, *my two cents* means *one's viewpoint*, rather than actual money. Increasing attention has been paid to idiom translation in recent years. Dankers et al. (2022b) has found that transformer encoders tend to process figurative idioms as single units more strongly, while Baziotis et al. (2023); Liu et al. (2023) have shown different training methods can be helpful for idiom translation.

However, much of this research has focused on idioms in English, Japanese, and European languages, with little attention given to Chinese idioms. From a linguistics perspective, the

| Error Type & Description | Example (zh) | Example (en) |
|---|---|---|
| **No Error:** Translation is of high quality. | ...科学家们欢呼雀跃，却发现... | ...Scientists cheered and rejoiced, but they discovered that... |
| | ...油霸们再捅刀美元，项庄舞剑意在沛公... | ...oil tycoons stab the US dollar again, Xiang Zhuang's sword dance is aimed at Pei Gong... |
| **Mistranslation:** Translation is incorrect and disrupts idiom understanding. | ...问得太多只会自寻烦恼... | Asking too many questions will only bring them problems. (*bring yourself problems*) |
| **Unnatural:** Translation is understandable yet suboptimal, and it can be improved. | ...善战者无赫赫之功... | A good warrior does not achieve a glaring victory... (*a remarkable victory*) |
| **Literal:** Translation is literal and coherent with the surrounding context. | ...这一生身无长处... | ...I have no merits in this life... (*live in poverty*) |
| **Addition:** Translation contains irrelevant content beyond the accurate meaning. | ...不仅于剧学素有深造，无所不通... | ...not only well-versed in drama studies, knowledgeable and passionate in all aspects... (*knowledgeable in all aspects*) |
| **Partial:** Given idiom is translated partially. | ...此处不留人，自有留人处，对于这类无良民宿... | ...there is no place to stay, and against these bad inns... (*if you don't stay here, there is a place to stay elsewhere*) |
| **Repetition:** The translation contains repeated correct content or synonyms. | ...最佳的父子关系应该如兄如弟... | ...the best relationship between a father and son should be like that of a brother and a brother... (*like that of brothers*) |
| **No Translation:** Translation lacks the idiom's meaning in the output. | ...正从布隆泉飞向普利托利亚，除了不速之客的蛇外，机上还载了4名乘客。 | ...was flying from Bloemfontein to Pretoria and had four other passengers on board, including the snake. (the *unwelcome* snake) |

Table 1: The IDIOMEVAL error taxonomy with example translations with each error type taken from our collected dataset in Section 3.1. The corrected span is included in *italics*. Participants can select More than One if there is more than one error present.

former falls into the Indo-European language family, while Chinese belongs to the Sino-Tibetan language family (Katzner & Miller, 2002). Besides, idioms across languages differ in their definitions and classifications. For example, while English idioms often emphasize syntactic structures, Chinese idioms place more importance on semantic and pragmatic aspects (Wang, 2021). Meanwhile, Chinese idioms generally contain proverbs, allegorical sayings, and aphorisms which mostly originate from ancient literature (Wang, 2021; 2022). One example is the idiom 明日黄花, which first appeared in the Song dynasty (ruled China from 960 to 1279 CE) and is literally translated into "tomorrow's yellow flower", but its figurative meaning is "things that have become outdated". Due to cultural and linguistic differences (Wang, 2022), Chinese idioms present unique challenges for translation systems.

In this work, we aim to shed light on how effectively modern translation systems handle Chinese idioms and whether current automatic metrics accurately capture the quality of idiom translations. We first introduce IDIOMEVAL, an error taxonomy capturing 7 different failure modes for Chinese idiom translation such as Mistranslation and Partial Translation (see Table 1. To explore the impact of domain context, we collect the Chinese texts that contain idioms from four distinct domains: Web, News, Wikipedia, and Social Media. Using the IDIOMEVAL taxonomy, we collect annotations for 900 translation pairs from 9 modern translation systems including GPT-4o (Achiam et al., 2023). With these annotations, we conduct an error analysis on system translations and investigate how well current evaluation metrics assess the quality of Chinese idiom translations. Our main findings are as follows:

- The selected systems struggle to produce high-quality idiom translations. They perform better on News but worse on Web and Social Media. The systems commonly produce Mistranslation, Literal Translation and Partial Translation.

- Current machine translation evaluation metrics do not correlate well with human annotations. They show moderate sensitivity to perturbed idiom translations and fail to detect idiom error spans reliably.

To improve the detection of good and bad idiom translations, we instruction-tune Qwen2.5 models and achieve a Macro $F_1$ score of 0.68, outperforming all existing metrics and prompting GPT-4o. We also find performance increases as the model scales up, except at 72B. We hope our work highlights the impact of idioms on translation and paves the way for developing better models and evaluation metrics. We will release all annotations, code, and models. In summary, our contributions are:

1. We introduce IDIOMEVAL, a new taxonomy of Chinese idiom translation errors.

2. We collect a high-quality dataset of 900 human-annotated translations spanning four domains and nine modern MT systems.

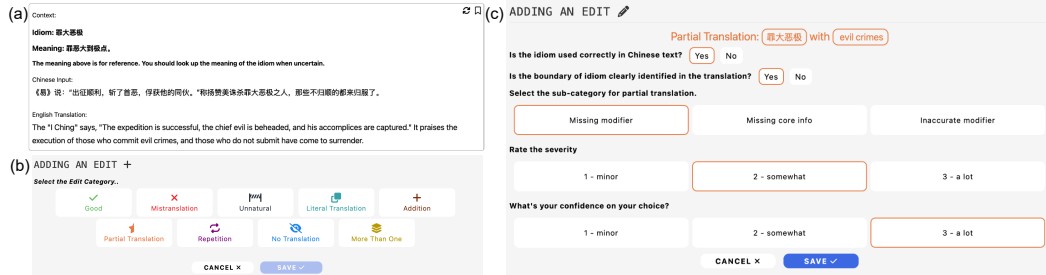

Figure 2: Idiom annotation framework.

3. We provide the first comprehensive analysis of how modern systems perform on idiom translation and how existing metrics correlate with human judgments.

4. We show that instruction-tuned LLMs significantly outperform existing metrics and prompting approaches in detecting idiom translation errors.

## 2 IDIOMEVAL: An Evaluation Framework

In this section, we discuss the details of our framework for idiom translation evaluation. IDIOMEVAL involves two steps: annotators select the translation span for the idiom within the context and classify it into 9 high-level linguistically grounded categories that are further broken down into 13 subcategories; annotators then rate the severity of the translation errors and provide their confidence in the annotations. Our annotation interface is built with Thresh (Heineman et al., 2023), and we will release our configuration for public use.

**Select Translation Span and Category.** The framework begins with selecting the corresponding translation span of the idiom. Annotators are then asked to choose one category that best describes the idiom translation. For certain categories, annotators are also required to select a subcategory. The IDIOMEVAL categories are defined in Table 1. We provide a detailed description of each category, subcategory and examples in Appendix A.

**Rate Severity and Confidence.** As each translation category can affect the overall sentence quality to varying degrees, we ask annotators to rate the severity of the errors. Following Karpinska et al. (2022); Dou et al. (2022), we define three levels of severity: minor (1), somewhat (2), and a lot (3). Finally, we ask annotators their confidence that their selection is correct on the same scale. An overview of the annotation interface is shown in Figure 2.

## 3 Data Collection and Annotation

We describe our methodology for automatically constructing a dataset of 623K Chinese sentences containing idioms across four domains. Using a subset of this data, we then collect and annotate 900 translations from 9 modern systems, analyzed in §4 and §5.

| Domain | Source | Period | Instances | Idioms | Avg. occurrence by freq. range | | | | |
|---|---|---|---|---|---|---|---|---|---|
| | | | | | VH | H | M | L | N |
| News | Common Crawl News | Mar.-Apr.,Oct | 50,845 | 5,333 | 16.7 | 3.9 | 2.5 | 1.5 | 1.5 |
| Web | Common Crawl | Jan.-Feb.,Dec. | 463,642 | 15,319 | 61.1 | 2.6 | 1.4 | 1.4 | 9.3 |
| Wikipedia | Wikipedia Meta History | Jul.,Dec. | 39,699 | 5,947 | 12.1 | 2.1 | 1.4 | 1.2 | 1.1 |
| Social Media | Weibo Search Results | Jan.-Jul.,Jan.-Dec. | 55,315 | 351 | 519.0 | 274.25 | 84.2 | 22.4 | 5.0 |

Table 2: Statistics of collected Chinese data by domain. *Source* lists the data collection sources. *Period* lists the month of collected data in 2023 and 2024. *Instances* shows the number of data samples in each domain. *Idiom* column shows the total number of idioms in the collected data. The rightmost column shows the average occurrence of idioms within each frequency range. The number of idioms within each frequency range can be found in Appendix B.2.

### 3.1 Collect Text with Idioms

We use the idiom vocabulary provided by Tan & Jiang (2021) for its high coverage, which contains 30,999 idioms with definitions. To evaluate idiom translations under different contexts, we first collect Chinese texts containing idioms from four domains: News, Web, Wikipedia, and Social Media. Our subsequent analysis involves LLMs that were released in 2023 and 2024. To minimize the chance that these texts appear in the pre-training corpora of LLMs used for translation (§3.2), we collect texts that were available after the respective release year of these LLMs (referred to as new corpus).

For each domain, we divide idioms into five groups according to the $1^{st}$, $2^{nd}$ and $3^{rd}$ quartiles of the overall frequency: VH (very high), H (high), M (medium), L (low), and N (never appear in the data). This allows us to measure the impact of idioms frequency in pre-training corpora. We use the old corpus (data before 2023 in the same domains, details in Appendix B.1) to calculate idiom frequencies, as idioms from this period are more likely to be included in the LLMs' training data. Given that idioms tend to have a long-standing presence in a language, their frequency distribution within the same domain is unlikely to change significantly over time. Table 2 displays statistics of the collected data. The details of data collection for each domain are described in Appendix C.1.

### 3.2 Collect Translations from Systems

We carefully select a diverse set of translation systems to allow us to assess idiom translation performance across a range of system types: GPT-4 (a SOTA LLM) (Achiam et al., 2023), Alma-13B (a fine-tuned translation model) (Xu et al., 2024), Qwen-14B-Chat (Bai et al., 2023) (a bilingual LLM for Chinese and English), mT0-13B (a multilingual LLM) (Muennighoff et al., 2022), Google Translate (a widely used commercial service), GPT-4o (a SOTA LLM) (Hurst et al., 2024), and QWen2.5 instruction-tuned models (multilingual LLMs with 7B, 14B and 72B parameters) (Yang et al., 2024a;b).

We sample 5 instances from each frequency range, resulting in 25 Chinese texts to be translated for each domain. In total, this produces 900 translation pairs from the nine systems. The authors, native speakers of both Chinese and English, manually translated each Chinese text into English. These translations are used as references in our analysis.

### 3.3 Human Annotation

We recruit native Chinese speakers fluent in English through Prolific. The annotation process has two phases: the pilot phase and the main phase. In the pilot phase, 20 participants are provided with annotation guidelines and a quiz. The top 5 scorers are selected for the main phase. In the main phase, each translation pair is assigned to three annotators, ensuring equal workload distribution. The final annotations are determined by a majority vote, with a manual resolution by the first author for any ties. More details on inter-annotator agreement and quality control are provided in Appendix C.2.

## 4 Translation System Evaluation

We conduct a detailed analysis of the idiom translation quality of the modern systems. We discuss how each system performs in each domain and their relative comparison. We then examine how domain and idiom frequency impact translation quality, followed by a discussion on the severity of translation errors. We present our results in Figure 3, which displays the ratio of different translation categories from each system in the four domains. Further analysis, such as commonly seen errors, subcategories selected, error distribution by frequency range, and average severity scores for each category, can be found in Appendix D. The following are the key findings:

**1. GPT-4 and Qwen family have overall higher translation quality, while Alma and mT0 have performed consistently badly.** In 2023 data, GPT-4 has the highest number of Good Translations across all the domains, while Qwen achieves a comparable second performance.

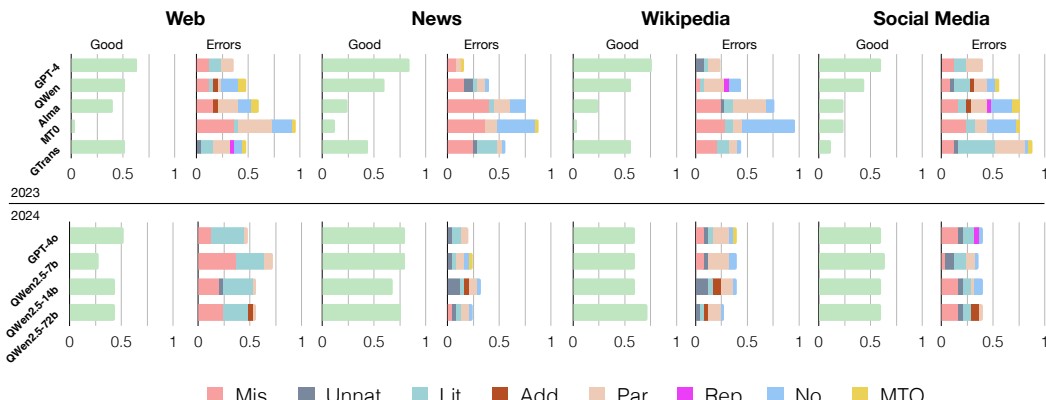

Figure 3: System-centric view of translation results in each domain. Each dataset is accompanied by two subfigures. Left subfigure: ratio of Good Translations. Right subfigure: composition of error categories for each system.

On News and Wikipedia, GPT-4 surpasses the second-best system (Qwen) by a margin of 0.24 and 0.2. The difference is minimized in the 2024 data, where both models have close quality. Despite the best performance among the models, the GPT-4 and Qwen family still fail to achieve satisfying results, with more than 20% of all translations being problematic. For Alma and mT0, fewer than 25% of their translations are rated as good (except for Alma on Web). This disparity is likely due to a lack of idiom-related Chinese data in the pre-training and fine-tuning phases. For example, Alma's backbone, LLaMA-2, is primarily pre-trained in English, limiting its performance. In contrast, Qwen, which is pre-trained on both Chinese and English bilingual data, achieves better translation quality.

**2. Systems' translation quality varies by domain, where they perform better on News but worse on Web and Social Media.** Although GPT-4 and Qwen provide the highest translation quality overall, their number of Good Translations varies significantly across domains. For GPT-4 and GPT-4o, more than 80% of its translations are rated as good on News, but this drops to 50% on Web and 60% on Social Media. The Qwen models have shown similar fluctuations. Overall, translation quality is worse on Web and Social Media. This may be because the two domains often feature newly emerging contexts, where idioms are used in novel ways compared to other domains. These variations highlight intrinsic differences in idiom usage across domains and emphasize the need for diverse training data to ensure consistently high-quality translations.

**3. Translation error types differ across systems, but Mistranslation and Partial Translation are commonly seen.** Errors made by GPT-4 and GPT-4o are mostly Mistranslation and Partial Translations. On Web, GPT-4o also produces a number of Literal Translations. Qwen (including Qwen2.5) and Alma produce more diverse errors. They are the only two models producing Addition errors (Web and Social Media). mT0 frequently produces No Translations, particularly on Wikipedia, where it fails to translate at least 40% of idioms across frequency ranges. Google Translate frequently produces Literal Translations (36% on Social Media) and Unnatural Translations across three domains (4%).

## 5 Automatic Metric Evaluation

We evaluate how well current automatic translation metrics assess idiom translations, focusing on their ability to evaluate translations containing idioms, their sensitivity to idiom translation errors, and whether they can detect specific idiom translation error spans.

### 5.1 Metric Performance

We consider six of the most widely-used and strongest automatic metrics (Freitag et al., 2023): BLEU (Papineni et al., 2002), BERTScore Zhang et al. (2019), COMET (Rei et al., 2022a), COMETKIWI (Rei et al., 2022b), MetricX-XXL (Juraska et al., 2023; 2024) and MetricX-QE-

| Scope | Category | Size | Reference-Based Metrics | | | | | Reference-Free Metrics | | |
|---|---|---|---|---|---|---|---|---|---|---|
| | | | BLEU | BERTScore | COMET | MetricX-23-XXL | MetricX-24-XXL | COMETKIWI | MetricX-QE-23-XXL | MetricX-QE-24-XXL |
| Full | Mistranslation | 128 | 0.071 | 0.119 | 0.170 | 0.118 | 0.170 | 0.215 | 0.080 | 0.140 |
| | Literal | 90 | 0.250 | 0.325 | 0.376 | 0.305 | 0.270 | 0.100 | 0.144 | 0.083 |
| | Partial | 105 | 0.254 | 0.208 | 0.284 | 0.163 | 0.244 | 0.136 | 0.322 | 0.341 |
| | No Translation | 74 | 0.186 | 0.271 | 0.348 | 0.198 | 0.226 | 0.081 | 0.138 | 0.268 |
| Idiom | Mistranslation | 128 | -0.074 | 0.056 | 0.239 | 0.251 | 0.166 | 0.174 | -0.021 | 0.055 |
| | Literal | 90 | 0.113 | 0.140 | 0.217 | 0.219 | 0.119 | -0.081 | -0.095 | -0.122 |
| | Partial | 105 | 0.131 | 0.109 | 0.107 | -0.112 | -0.261 | -0.034 | -0.332 | -0.326 |
| | No Translation | 74 | * | * | -0.052 | -0.025 | -0.512 | -0.088 | -0.502 | -0.536 |
| Full | All errors | 452 | 0.277 | 0.322 | 0.376 | 0.264 | 0.241 | 0.208 | 0.186 | 0.175 |
| | All categories | 900 | 0.274 | 0.352 | 0.386 | 0.341 | 0.292 | 0.151 | 0.131 | 0.090 |
| Idiom | All errors | 452 | 0.227 | 0.229 | 0.343 | 0.250 | -0.122 | -0.004 | -0.277 | -0.274 |
| | All categories | 900 | 0.343 | 0.399 | 0.473 | 0.483 | 0.036 | 0.026 | -0.161 | -0.205 |

Table 3: Pearson's *r* between metrics and human annotations. Evaluation is measured on both *Full* translation and *Idiom* translation. More Than One is omitted here as all its instances are rated as the highest severity. Unnatural, Addition and Repetition are omitted due to small sample size. *: for No Translation on *Idiom*, BLEU and BERTScore outputs 0 and thus omitted. Kendall's $\tau$ results (Table 9 in Appendix E.1) show similar patterns.

| Metric | ROC-AUC | | Good > Bad | |
|---|---|---|---|---|
| | Full | Idiom | Full | Idiom |
| BLEU | 0.63 | 0.69 | 68% | 54% |
| BERTScore | 0.67 | 0.71 | 72% | 77% |
| COMET | 0.67 | 0.75 | 64% | 56% |
| MetricX-23-XXL | 0.67 | 0.78 | 67% | 78% |
| MetricX-24-XXL | 0.66 | 0.57 | 71% | 76% |
| COMETKIWI | 0.58 | 0.51 | 58% | 51% |
| MetricX-QE-23-XXL | 0.58 | 0.47 | 65% | 60% |
| MetricX-QE-24-XXL | 0.59 | 0.45 | 64% | 62% |

| Perturbation | COMETKIWI | | MetricX-QE-23-XXL | | MetricX-QE-24-XXL | |
|---|---|---|---|---|---|---|
| | Full | Idiom | Full | Idiom | Full | Idiom |
| Mistranslation | 0.73 | 0.66 | 0.80 | 0.62 | 0.79 | 0.67 |
| Unnatural | 0.59 | 0.57 | 0.56 | 0.52 | 0.66 | 0.52 |
| Literal | 0.65 | 0.65 | 0.73 | 0.76 | 0.72 | 0.73 |
| Addition | 0.66 | 0.57 | 0.54 | 0.52 | 0.72 | 0.48 |
| Partial | 0.59 | 0.58 | 0.61 | 0.57 | 0.72 | 0.65 |
| Repetition | 0.67 | 0.54 | 0.74 | 0.71 | 0.72 | 0.69 |
| No Translation | 0.68 | 0.28 | 0.75 | 0.75 | 0.72 | 0.77 |

Table 4: Performance of existing metrics on idiom translation. Left table: distinguishing good and bad translations based on ROC-AUC scores and pairwise comparison accuracy on *Full* text and *Idiom* translations. Right table: sensitivity of reference-free metrics on *Full* text and *Idiom* translations, measured by the ratio of instances where unperturbed translations score better than the perturbed translations.

XXL (Juraska et al., 2023; 2024). Among these, COMETKIWI and MetricX-QE are reference-free. To allow for comparison with human annotations, we collapse each annotation into a single numeric score, which is the severity score (from 1 to 3) directly as the overall score. Good translations are marked as 0.

To understand how well these metrics evaluate idioms both in context and individually, we calculate their scores based on the full context and only the idiom part. Table 3 reports the Pearson correlation coefficient (*r*) between the metrics and human evaluation for different error types. We found that, at both the full context and idiom levels, the metrics struggle to provide effective fine-grained measurements, with the highest correlation being 0.386 for full context and 0.483 for idioms. In most cases, correlations are stronger when measured on idiom translations than in full text, as the surrounding context of idioms also influences the metric scores. Nevertheless, existing metrics show weak correlations with human ratings, thus failing to measure the quality of idiom translation from a human-like perspective. Similar trends are observed when examining the ranking correlation with Kendall's $\tau$ (details in Appendix E.1).

Given that the metrics cannot effectively measure the severity of errors, we next investigate a simpler question: Can they distinguish good idiom translations from bad ones? We report the ROC-AUC score of each metric and the percentage of pairs where good translations score better than bad ones (for the same Chinese sentence) in Table 4 (left). BERTScore and MetricX perform the best on both metrics. However, in more than 20% of cases, Good Translations are still rated lower than those with errors, indicating that these metrics are not consistently reliable for distinguishing between good and bad idiom translations.

## 5.2 Metric Sensitivity

We measure metric sensitivity based on a similar approach to Karpinska et al. (2022) by comparing how each metric scores gold translations versus perturbed translations. Metrics that are sensitive to perturbations can accurately identify the perturbations. We focus on reference-free metrics, COMETKIWI and MetricX-QE as reference-based metrics use the gold translation as reference. To obtain perturbed translations, we use GPT-4o (OpenAI, 2024) to edit the existing reference translations into different errors. The task is decomposed into two steps: 1) Extraction: extracting the idiom translation from the English sentence, and 2) Perturbation: perturbing idiom translations with errors (see Appendix E.2 for the prompts). For each error type, we randomly sampled 10 instances and checked if the edited translations matched the intended error category. More than 70% of the instances were edited correctly for each category. Since the model was specifically instructed to introduce errors, the gold translation should always be better than the perturbed ones.

We calculate the accuracy on perturbation as the ratio of instances where gold translations score better than perturbed translations, as shown in Table 4 (right). For full-context translations, the top-performing metric, MetricX-QE-24, has an accuracy of around 0.7 under most perturbations. In contrast, COMETKIWI and MetricX-QE-23 have an accuracy below 0.6 for certain perturbations. However, when evaluating translations on the idiom part, the metrics demonstrate lower performance in almost all perturbations, showing they are less effective at detecting errors when only the idiom is presented.

## 5.3 Can Existing Metrics Identify Idiom Error Span?

We use xCOMET Guerreiro et al. (2023), the state-of-the-art error span detection model, to examine if it can accurately detect the span of idiom translation errors. We evaluate the performance using character level $F_1$ score and only consider target spans. Similar to Blain et al. (2023), we also compute weighted $F_1$ scores by converting severity scores into binary weights. Table 5 displays the corresponding performance. We find that xCOMET performs poorly in detecting the span of idiom translation error, with the best model achieving only 0.3 $F_1$.

|  | P | R | $F_1$ | WP | WR | W$F_1$ |
|---|---|---|---|---|---|---|
| xCOMET-XL | 12.5 | 44.7 | 16.6 | 0.4 | 2.2 | 0.7 |
| xCOMET-XXL | 10.2 | 30.3 | 12.4 | 0.5 | 1.6 | 0.6 |
| xCOMET-XL (src) | 13.3 | 49.7 | 18.1 | 0.5 | 2.4 | 0.7 |
| xCOMET-XXL (src) | 9.3 | 11.4 | 30.0 | 0.5 | 1.6 | 0.5 |

Table 5: Precision and $F_1$ scores for error span detection using xCOMET. *src* means reference-free version. *W* represents weighted metric output. All values are multiplied by 100 for readability.

# 6 Improving Detection of Good and Bad Idiom Translation

With our corpus, we aim to develop a model that determines whether a Chinese idiom is correctly translated. The task is formulated as a binary classification problem: let $i$ be the idiom, $s$ be the source Chinese text containing $i$, and $t$ be the English translation of $s$. We want to predict whether $i$ is correctly translated in $t$. In our experiments, we only use references for reference-based metrics, such as BLEU. All other methods remain reference-free since references may be unavailable in real-world scenarios.

## 6.1 Data and Baselines

In practice, metrics are used to evaluate newly released models, so we split our data by model: the training and validation sets include translations from older models (GPT-4, Qwen, Alma, mT0, and Google Translate), while the test set includes translations from newer models (GPT-4o, Qwen2.5-7B, Qwen2.5-14B, and Qwen2.5-72B), leading to different sampled idioms with minimum overlap. The train, validation, and test sets contain 450, 50, and 400 instances, respectively.

We consider two baselines: regression-based metrics and LLM-based prompting. The regression metrics produce a continuous score, so we use the training set to find an optimal

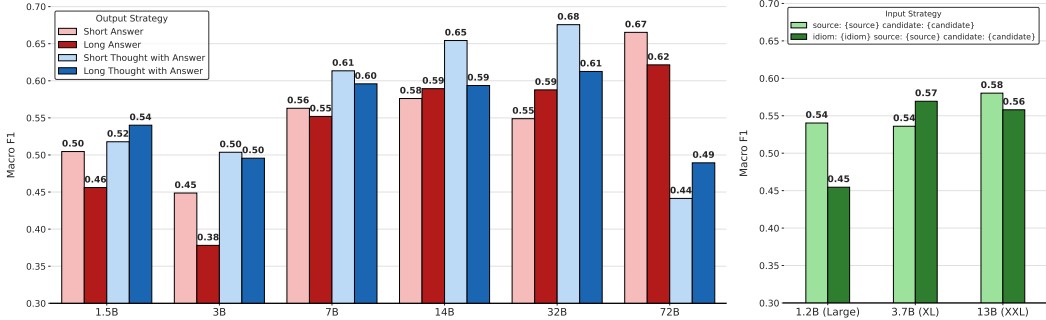

Figure 4: Performance comparison across model sizes and strategies for training Qwen2.5 (left) and MetricX (right). For Qwen2.5 (1.5B–32B), a short thought plus final answer yields the best results, while at 72B, a shorter answer format performs similarly. For MetricX, omitting the idiom gives better results at 1.2B and 13B. Overall, scaling model sizes improves performance, except at Qwen2.5-72B.

threshold from the ROC curve, then apply it to the test set. For prompting, we evaluate GPT-4o and Qwen2.5 with zero-shot direct-answer and CoT prompts (Wei et al., 2022).

## 6.2 Our Methods

We explore two fine-tuning approaches. First, we fine-tune MetricX-24, the top-performing machine translation metric in WMT24 (Freitag et al., 2024). Second, we instruction-tune the Qwen2.5 family of models, the best open-source LLMs in Chinese and English.

**Fine-tuning MetricX.** MetricX is based on mT5 (Xue et al., 2020) and trained to predict a regression score from 0 to 25 via the logits for a special token from the LM head. To adapt it for binary classification, we replace the LM head with a classification head that inputs the decoder's final-layer representation of the starting token. We compare two input formats for training: one that includes the idiom ("idiom: {idiom} source: {source} candidate: {candidate}") and another that omits it ("source: {source} candidate: {candidate}"), which follows the original MetricX-24 training format. This allows us to study how adding extra information interacts with the existing format in a limited training data setting.

**Instruction tuning Qwen2.5.** We format each instance as an instruction, where the input includes the idiom, source Chinese text, and English translation. For output, on which the model's loss is computed, we test four strategies: (1) a simple yes/no answer,

| Model | #Params. | $F_1$-Good | $F_1$-Bad | $F_1$-Macro |
|---|---|---|---|---|
| **Dummy** | ∼ | 0.76 | 0.00 | 0.38 |
| | Thresholded Metrics Based on Roc Curve | | | |
| *Reference-Based* | | | | |
| **BLEU** | ∼ | 0.68 | 0.43 | 0.56 |
| **BERTScore** | 355M | 0.72 | 0.43 | 0.57 |
| **COMET** | 580M | 0.65 | 0.50 | 0.58* |
| **MetricX-23** | 13B | 0.68 | 0.54 | 0.61* |
| **MetricX-24** | 13B | 0.66 | 0.54 | 0.60* |
| *Reference-Free* | | | | |
| **COMETKIWI** | 3.5B | 0.49 | 0.52 | 0.50 |
| **MetricX-QE-23** | 13B | 0.60 | 0.37 | 0.49 |
| **MetricX-QE-24** | 13B | 0.63 | 0.29 | 0.46 |
| | Prompting LLMs | | | |
| *Zero-shot* | | | | |
| **Qwen2.5** | 14B | 0.65 | 0.49 | 0.57 |
| **Qwen2.5** | 72B | 0.75 | 0.37 | 0.56 |
| **GPT-4o** | ∼ | 0.74 | 0.51 | 0.63 |
| *Zero-shot-CoT* | | | | |
| **Qwen2.5** | 14B | 0.73 | 0.47 | 0.60* |
| **Qwen2.5** | 72B | **0.78** | 0.30 | 0.54 |
| **GPT-4o** | ∼ | **0.78** | 0.42 | 0.60* |
| | Fine-tuned Models | | | |
| *Fine-tuning Encoder-Decoder with Classification Head* | | | | |
| **MetricX-QE-24** | 13B | 0.66 | 0.50 | 0.58 |
| *Instruction Tuning Decoder-Only LLMs* | | | | |
| **Qwen2.5** | 14B | 0.77 | 0.54 | 0.65 |
| **Qwen2.5** | 72B | 0.70 | **0.63** | **0.67** |

Table 6: Comparison of our fine-tuning methods with threshold-based metrics and LLM prompting. Fine-tuned models achieve higher performance than the baselines.

(2) a slightly longer answer, (3) a short thought process followed by the final answer, and (4) a longer thought process followed by the final answer. The latter three strategies use templates described in Appendix F, where we also provide additional details for all methods.

### 6.3 Results

Table 6 shows the $F_1$ scores for both good and bad translation classes, as well as their Macro average. We conduct one-sided paired bootstrap significance tests (10,000 resamples) to compare each metric's macro $F_1$ score against that of the best-performing model, fine-tuned Qwen2.5-72B. An asterisk (*) indicates statistical significance ($p < 0.05$). We show only the top-performing metrics from each method category. Figure 4 provides a more detailed comparison across different model sizes and training strategies for fine-tuning MetricX and instruction-tuning Qwen2.5.

**Instruction tuning Qwen2.5 outperforms all baselines, followed by prompting GPT-4o and then thresholded reference-based metrics.** Fine-tuning Qwen2.5-32B with a short thought process and final answer achieves the highest Macro $F_1$ of 0.68. Fine-tuning MetricX-QE-24 raises its Macro $F_1$ from 0.46 to 0.58. For prompting, GPT-4o reaches a macro $F_1$ of 0.63, while the Qwen2.5-72B gets 0.56. Interestingly, chain-of-thought prompting lowers performance on both of these two models, showing these models are more inclined to judge translations as correct when CoT is used. For thresholded metrics, reference-based metrics outperform reference-free ones by a significant margin, with simple BLEU outperforming COMETKIWI by 0.06 $F_1$. **Model size scaling generally helps Qwen2.5 and MetricX, with an exception at 72B.** Across the 1.5B to 32B versions of Qwen2.5, using a short thought process plus the final answer yields the best results. However, at 72B, training on short answer performs the best, on par with the best 32B variant. For MetricX, using its original training format (omitting the idiom) works better at the 1.2B and 13B scales, indicating that data format consistency might be crucial when training data is limited.

We further investigate why the performance does not scale up at Qwen2.5 72B. In terms of training loss (cross-entropy), we found that when fine-tuned on thoughts, the 72B model plateaus around 0.2 to 0.3, while smaller models (32B, 14B, 7B) converge more smoothly to a lower loss ($\tilde{0}.02$). In contrast, training the 72B model on answers only, especially short answers, results in lower loss and better performance.

In terms of model behavior, we find that the 72B model always outputs the short answer format (yes / no) even when fine-tuned on longer formats such as long answers or thought-based responses. This differs from all the smaller size models (1.5B to 32B), which follow the training output format. While we do not fully understand the root cause of this behavior, it may explain why training the 72B model on the short answer produces the best performance.

## 7 Related Work

**Machine Translation with Idioms.** We consider multi-word expression (MWE) a close line of work to idioms, and there have been plenty of studies on MWEs (Sag et al., 2002; Calzolari et al., 2002). Zaninello & Birch (2020) explored annotation and data augmentation techniques for MWE translations. Fadaee et al. (2018) add additional input features by pretending special tokens to indicate the presence of idioms. Gamallo & Garcia (2019) proposed compositional translation using cross-lingual embeddings. More recent works focus on idiom identification (Tedeschi et al., 2022), characterization (Socolof et al., 2021) and representation (Zeng & Bhat, 2022). Dankers et al. (2022b); Baziotis et al. (2023) provide an analysis of idiom processing with Transformers in the context of neural machine translation. Liu et al. (2023) provide a characterization of idiomatic translation and its issues and introduce upweighting and a retrieval module to improve the translation. In contrast to these works, our research introduces a fine-grained evaluation framework for idiom translation and highlights the limitations of modern LLMs, such as GPT-4, and automatic metrics in generating and evaluating idiom translations.

**Studies with Chinese Idioms.** There has been limited research involving Chinese idioms. Zheng et al. (2019) proposed a large-scale Chinese idiom dataset ChID for cloze tests. It is the first work to study Chinese idiom understanding in the form of machine reading comprehension. Tan & Jiang (2021) constructs a dataset to facilitate the evaluation of Chinese idiom embeddings. Tan et al. (2021) collected a large pretaining corpus of Chinese text by

crawling e-books online. Liao & Cheng (2023) studied text polishing with Chinese idioms, where sentences are rewritten to be more elegant with idioms. Other works have focused on creating parallel corpus with Chinese idioms. Shao et al. (2018) constructed the CIBB dataset that contains the English translations of 50 distinct Chinese idioms, together with a blacklist of literal translations. Wang & Yu (2010) created CIKB dataset that contains about 38k idioms and different English translations. However, only 28% of all samples have complete translations, and the dataset is not publicly available. Tang (2022) constructed PETCI dataset that contains idioms collected from a dictionary, together with translation results from Google and DeepL. More recently, Tang et al. (2024) used GPT-4 to generate context-aware idiom translations, yet there is a lack of external datasets and expert evaluations.

## 8    Conclusion

We introduce IDIOMEVAL, a framework with a carefully designed taxonomy that facilitates evaluating Chinese idiom translation. We posit that our framework can be extended to different languages. Our evaluation of various systems across multiple domains reveals that they struggle to produce high-quality idiom translations. We also find that existing metrics fail to correlate with human ratings or reliably distinguish good translations from poor ones. Furthermore, we demonstrate that the fine-tuned Qwen2.5 model is able to identify good and bad idiom translations with a Macro $F_1$ of 0.68, surpassing existing metrics and prompting-based LLMs. Our work lays the groundwork for future research to improve idiom translation performance and to develop metrics focused on idioms.

## Acknowledgement

This research is supported in part by the NSF CAREER award IIS-2144493. The views and conclusions contained herein are those of the authors and should not be interpreted as necessarily representing the official policies, either expressed or implied, of NSF or the U.S. Government. The U.S. Government is authorized to reproduce and distribute reprints for governmental purposes notwithstanding any copyright annotation therein.

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

# A Idiom Translation Examples

Below are the input and output of different translation categories and subcategories, along with a corresponding explanation. We skip trivial cases.

**No Error: Correct Underlying Meaning**

**Chinese**: 主断层绕过核电站，停了下来。美国地质调查局的科学家们欢呼雀跃，却发现断层并未被完全阻止。

**English**: The main fault bypassed the nuclear power plant and came to a stop. Scientists from the United States Geological Survey cheered and rejoiced, but they discovered that the fault had not been completely halted.

**Explanation**: 欢呼雀跃 is translated correctly into "cheered and rejoiced".

**No Error: Literal and Coherent**

**Chinese**: 此举指向性明确——挑衅美元，直接让联储局骑虎难下。（更多内容见：《沙特为首的油霸们再捅刀美元项庄舞剑意在沛公》）3月，沙特一连串的'出牌'，是沙特战略转型的重要一步。

**English**: This move has a clear direction - provoking the US dollar and putting the Federal Reserve in a difficult position. (For more content, see: "Saudi-led oil tycoons stab the US dollar again, Xiang Zhuang's sword dance is aimed at Pei Gong") In March, Saudi Arabia's series of 'moves' is an important step in its strategic transformation.

**Explanation**: The original meaning of 项庄舞剑 means the actual intention is not what it appears to be. However, in this case, the idiom is used to refer to the historical story and a literal translation is more appropriate.

**Mistranslation**

**Chinese**: 另一半有时需要一个人安静地思考，此时你不必过多打扰，问得太多只会自寻烦恼。

**English**: The other half sometimes needs someone to think quietly by themselves, at which point you don't need to disturb them too much. Asking too many questions will only bring them problems.

**Explanation**: 自寻烦恼 means asking for trouble or trouble someone him/herself, not "bringing other people trouble".

**Unnatural**

**Chinese**: 善战者无赫赫之功。

**English**: A good warrior does not achieve a glaring victory.

**Explanation**: 赫赫之功 means "a great achievement" or "a great victory". "A glaring victory" conveys a rough meaning but the usage of "glaring" is not perfect enough.

**Literal Translation**

**Chinese**: 我爸临终时候说，这一生身无长处，不曾留下什么给我

**English**: My father said when he was dying, 'I have no merits in this life, I have not left anything for you.'

**Explanation**: 身无长处 is translated literally here. 身: in this life. 无: no. 长处: merits.

**Addition**

**Chinese**: 程长庚不仅"于剧学素有深造，无所不通"，而且怀有崇高的戏剧理想和抱负。

**English**: Cheng was not only "well-versed in drama studies, knowledgeable and passionate in all aspects", but also harbored lofty theatrical ideals and ambitions.

**Explanation**: 无所不通  means knowledgeable in all aspects, and it is not related to one's passion.

**Partial Translation: Missing Modifier**

**Chinese**: 《易》说："出征顺利，斩了首恶，俘获他的同伙。"称扬赞美诛杀罪大恶极之人，那些不归顺的都来归服了。

**English**: The "I Ching" says, "The expedition is successful, the chief evil is beheaded, and his accomplices are captured." It praises the execution of those who commit evil crimes, and those who do not submit have come to surrender.

**Explanation**: 罪大恶极  refers to extremely evil crimes. Here, the translation has conveyed the core idea of the idea, but the degree of "evil" is missing.

**Partial Translation: Missing Core Info**

**Chinese**: 再者，平台繁多，此处不留人，自有 留人处，对于这类无良民宿，仅靠平台按现行规则管制，并不足以纠偏。

**English**: Also, there are many agencies, there is no place to stay, and against these bad inns, only the agencies with the current rules cannot adequately curb the abuse.

**Explanation**: Only the first half of 此处不留人，自有留人处  is translated. The second half of the idiom contains important information for us to understand the idiom, too.

**Partial Translation: Inaccurate Modifier**

**Chinese**: 被俘俄军惊恐万状，其实他比地上的同伙幸运多了。

**English**: The captured Russian soldiers are a bit terrified, but in reality, they are much luckier than their comrades on the ground.

**Explanation**: 惊恐万状  means "very terrified". The translation inaccurately used "a bit" as a modifier.

**Repetition**

**Chinese**: 我认为最佳的父子关系应该如兄如 弟。

**English**: I think the best relationship between a father and son should be like that of a brother and a brother.

**Explanation**: 如兄如弟  means the relationship is like brothers. In this case, the translation is repeated.

**No Translation: Missing from Output**

**Chinese**: 伊拉斯谟当时正从布隆泉飞向普利托利亚，除了不速之客的蛇外，机上还载了4名乘客。

**English**: Irascible was flying from Bloemfontein to Pretoria and had four other passengers on board, including the snake.

**Explanation**: 不速之客  is not translated and is missing from the English sentences.

**No Translation: Chinese Idioms Copied or Paraphrased**

**Chinese**: 他指平日儿子和媳妇都早出晚归，只有他们两老和两名孙女在家，他担心对方会变本加厉。

**English**: He指平日儿子和媳妇都早出晚归，只有他们两老和两名孙女在家，他担心对方会变本加厉。

**Explanation**: 早出晚归 is not translated and remains in Chinese text.

## B  Statistics of Corpus

### B.1  Statistics of Old Corpus

Table 7 shows the statistics of the old corpus.

| Domain | Source | Period | Instances | Idioms | Avg. occurrence by freq. range | | | |
|---|---|---|---|---|---|---|---|---|
| | | | | | VH | H | M | L |
| News | News Crawl | 2019 | 17,931,873 | 10,830 | 425.6 | 38.6 | 8.0 | 1.5 |
| Web | mc4 | 2008-22 | 54,542,308 | 30,652 | 8135.5 | 220.0 | 53.0 | 26.0 |
| Wikipedia | wiki2019zh | 2018-19 | 1,043,224 | 11,754 | 95.1 | 13.0 | 3.4 | 1.0 |
| Social Media | PchatbotW | 2008-22 | 139,448,339 | 15,089 | 2313.1 | 189.9 | 16.2 | 1.8 |

Table 7: Statistics of *old corpus*. *Source* lists the data collection sources. *Period* lists the year of collected data. *Instances* shows the number of data samples in each domain. *Idiom* column shows the total number of idioms. The rightmost column shows the average occurrence of idioms within each frequency range.

### B.2  Number of Idioms within Each Frequency Range

Table 8 displays the number of idioms in each frequency range in our newly collected data.

| Domain | VH | H | M | L | N |
|---|---|---|---|---|---|
| News | 2,486 | 1,768 | 692 | 225 | 162 |
| Web | 7,310 | 4,344 | 2,117 | 1,442 | 106 |
| Wikipedia | 2,814 | 1,702 | 772 | 269 | 390 |
| Social Media | 60 | 60 | 60 | 104 | 67 |

Table 8: Number of idioms within each frequency range in our collected data.

## C  Data Collection and Annotation Details

### C.1  Data Collection for Each Domain

**News.** We collect data from Common Crawl News (2023-04 and 2024-10 snapshots), consisting of 41,354 Chinese news articles, which we converted to simplified Chinese for consistency. For each article, we extracted sentences containing idioms, along with the preceding and following sentences for context, resulting in a total of 50,845 instances.

**Web.** We use Common Crawl (2023-06 and 2024-12 snapshots) to collect web page data. After retrieving the HTML content from each URL, we extracted sentences containing idioms along with their surrounding context, similar to the news data. This resulted in a total of 463,642 instances.

**Wikipedia.** We select the Chinese Wikipedia meta history (2023-07 and 2024-12 snapshots) for data collection. We save the revision IDs of articles whose first revision is after the cutoff

date. We obtain HTML pages for each revision through Wikimedia REST API[1]. We then extract sentences with idioms and their contexts from the revised text. There are 39,699 instances in total.

**Social Media.** We use Weibo, a popular Chinese social media platform similar to Twitter, to collect posts containing idioms. We rely on Weibo's search function to collect posts containing idioms. We check the publishing date of each post to ensure it was published after 2023. Unlike the previous domains, where we collected new data with all present idioms from the `old corpus`, querying each idiom will incur a high overhead in API usage. We instead sample 400 idioms from `old corpus` based on frequency range (60 for VH, H, M and 110 for L and N). We double the size for L and N to alleviate the scarcity of data. We managed to collect posts for 351 out of 400 idioms, and there are 55,315 instances in total.

## C.2 Quality Control

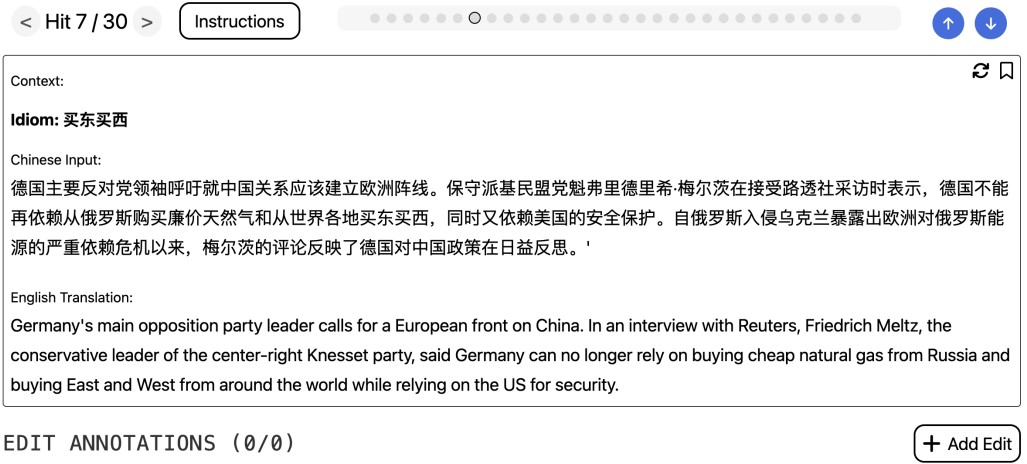

Figure 5: Example quiz question.

We use Prolific to recruit annotators who are native Chinese speakers fluent in English. We only collect the annotations on the data we provide, and no information about the annotators themselves is collected. We have received full consent from the annotators on this task. The annotation process is divided into two phases: *pilot phase* and *main phase*. In the pilot phase, 20 participants are selected for pre-screening. They are provided with detailed guidelines and must complete a quiz to demonstrate their understanding. The quiz consists of 30 questions, covering each category and subcategory with 2-3 questions. An example quiz question is shown in Figure 5. The top 5 performers from the pilot phase are selected for the main phase. In the main phase, we randomly assign each translation pair to three annotators while we ensure each annotator sees an equal number of pairs. We sample 100 annotations and evaluate them against the authors' own annotations, achieving a category accuracy of 66% and subcategory accuracy of 88%. For the final annotations, we take the majority vote and the first author manually resolves any tie cases. This resolution improves the average Cohen's $\kappa$ among annotators on category (subcategory) agreement from 0.47 (0.38) to 0.73 (0.69), indicating strong annotator agreement given some translations may have more than one answer. For instance, translating 虚情假意 (false feelings and intentions, indicating insincerity in one's intention) into "false" can be considered as either a Mistranslation or a Partial Translation. Annotators are compensated at $22 per hour.

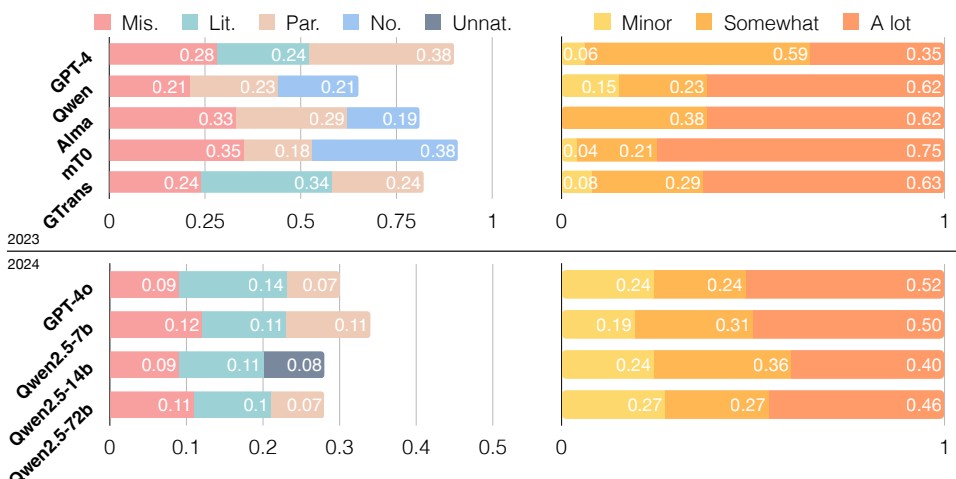

Figure 6: Left: Top three most commonly seen errors for each system. Mis.: Mistranslation; Lit.: Literal Translation; Par.: Partial Translation; No.: No Translation. Right: Ratio of each severity score for each system, calculated overall all errors and domains.

# D  Further Analysis on Translation Systems

## D.1  Category Breakdown

Figure 7 shows the subcategory composition for Good Translation, Mistranslation, and Partial Translation. It can be seen that subcategories vary across domains and systems. For instance, for Partial Translations, On News and Wikipedia, Alma is likely to omit modifiers (100% and 75%), while on Web and Social Media, it misses the core meaning in most cases (100% and 67%). mT0 is accomplished by a significant number of No Translations in each domain, where its translations are mostly missing from the output.

## D.2  Frequency Breakdown

Figure 9 illustrates each system's performance in each frequency range across domains. It can be seen that the frequency ranges of idioms are not correlated with translation quality. For idioms that appear in old corpus, their translation quality does not decrease with their frequency ranges. For instance, idioms in Web from the M range (100% Good Translations) get the best translation by GPT-4, while those from the H and L range (60% Good Translations) achieve comparable translation quality. Similar patterns can be found in other models. Interestingly, idioms that never appeared in the old corpus (N range) do not always result in poor translations. For instance, on Wikipedia, Alma produces the highest-quality translations for N-range idioms over other frequencies. This observation is particularly remarkable on Social Media data, where idioms that never appear get the best translation quality by almost all models in 2023 data. Such idioms get relatively good translation quality in 2024 data, too. Since the N range is defined regarding the appearance of idioms in old corpus for that domain. Idioms not showing up in one domain can still appear in another domain and be included in the pre-training corpus. This shows that language models transfer the knowledge from a different domain and translate the idiom correctly.

## D.3  Common Errors and Severity Scores

Figure 6 displays the top three most commonly seen errors for each system and the ratio of each severity score for each system. Figure 8 shows the breakdown of the severity score of each system on each error category across domains. We note that errors that interrupt

---

[1]https://zh.wikipedia.org/api/rest_v1/

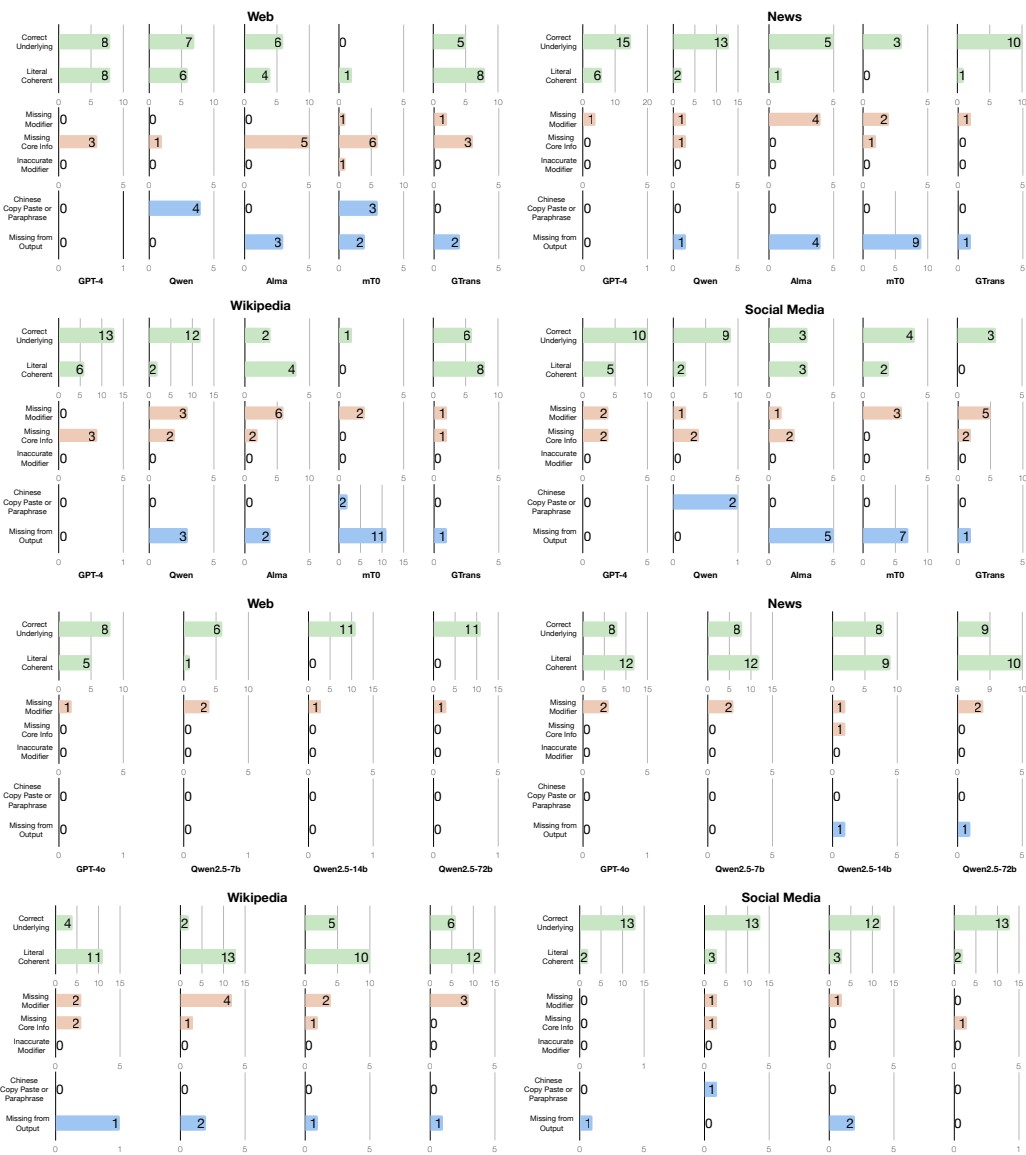

Figure 7: Subcategory composition for Good Translation, Partial Translation and No Translation.

meaning are more severe than errors that preserve the overall meaning. When models make errors, over half of them significantly impact the understanding of the text. For 2023 data, despite GPT-4's top performance, only 6% of its incorrect translations have a minor impact, placing it third behind Qwen and Google Translate. Alma makes the most severe errors, with none of its errors classified as minor. All the models have more than 40% of their errors marked as the highest severity in 2024 data.

Mistranslation and No Translation are the most severe error types, as they disrupt meaning significantly. Except for Wikipedia in 2024 data, Mistranslation has a minimum average severity score of 2.43. Meanwhile, No Translation has a severity score of 3 in many domains. In contrast, Unnatural Translation (max severity of 2), Addition (max severity of 2.5), and Repetition (max severity of 2) are less severe, as they typically preserve some correct or near-correct idiom parts.

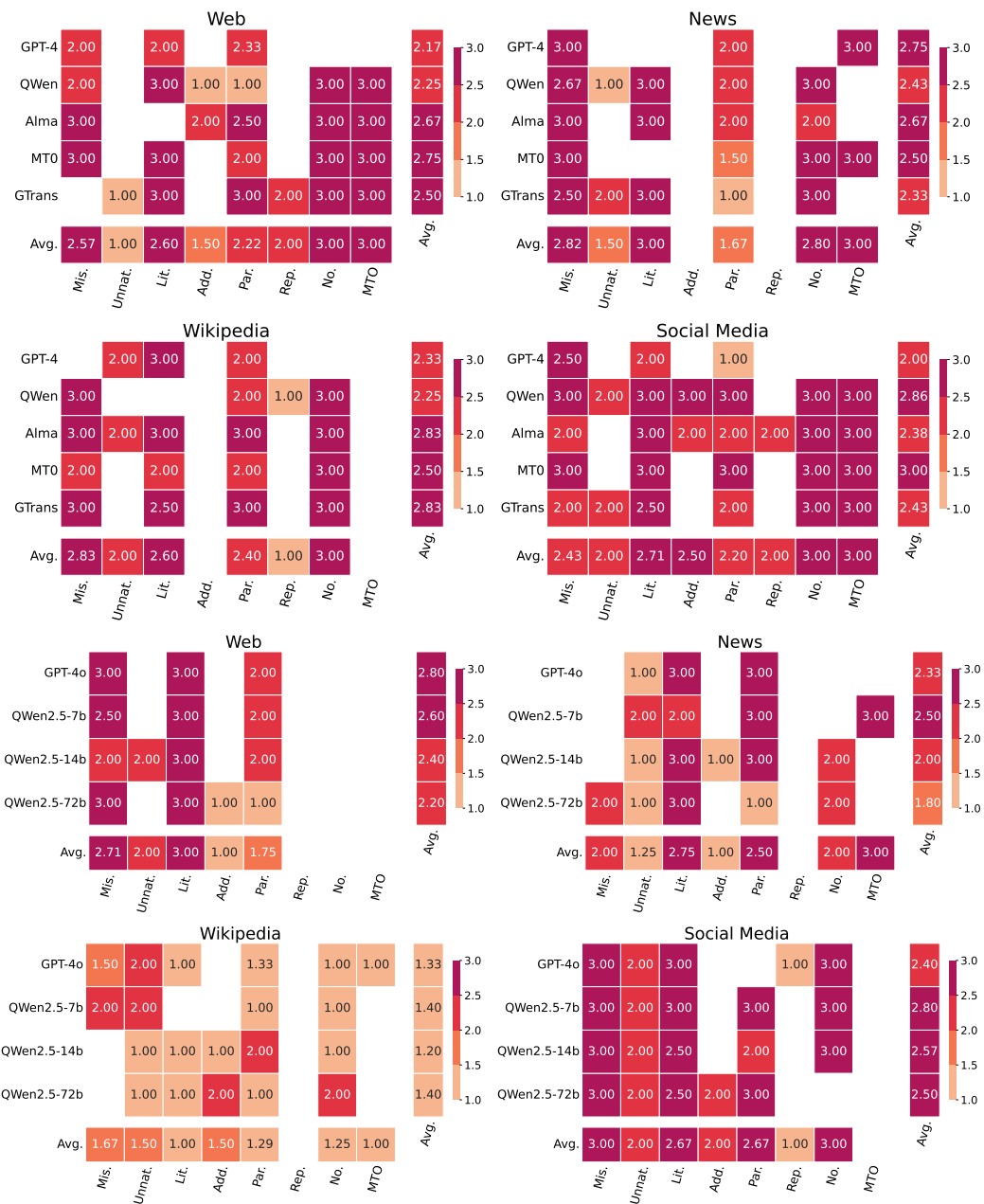

Figure 8: Average severity scores for each translation category in each domain. The right-most column represents the average severity score across all errors for each system. The bottom row represents the average severity score across all systems for each error.

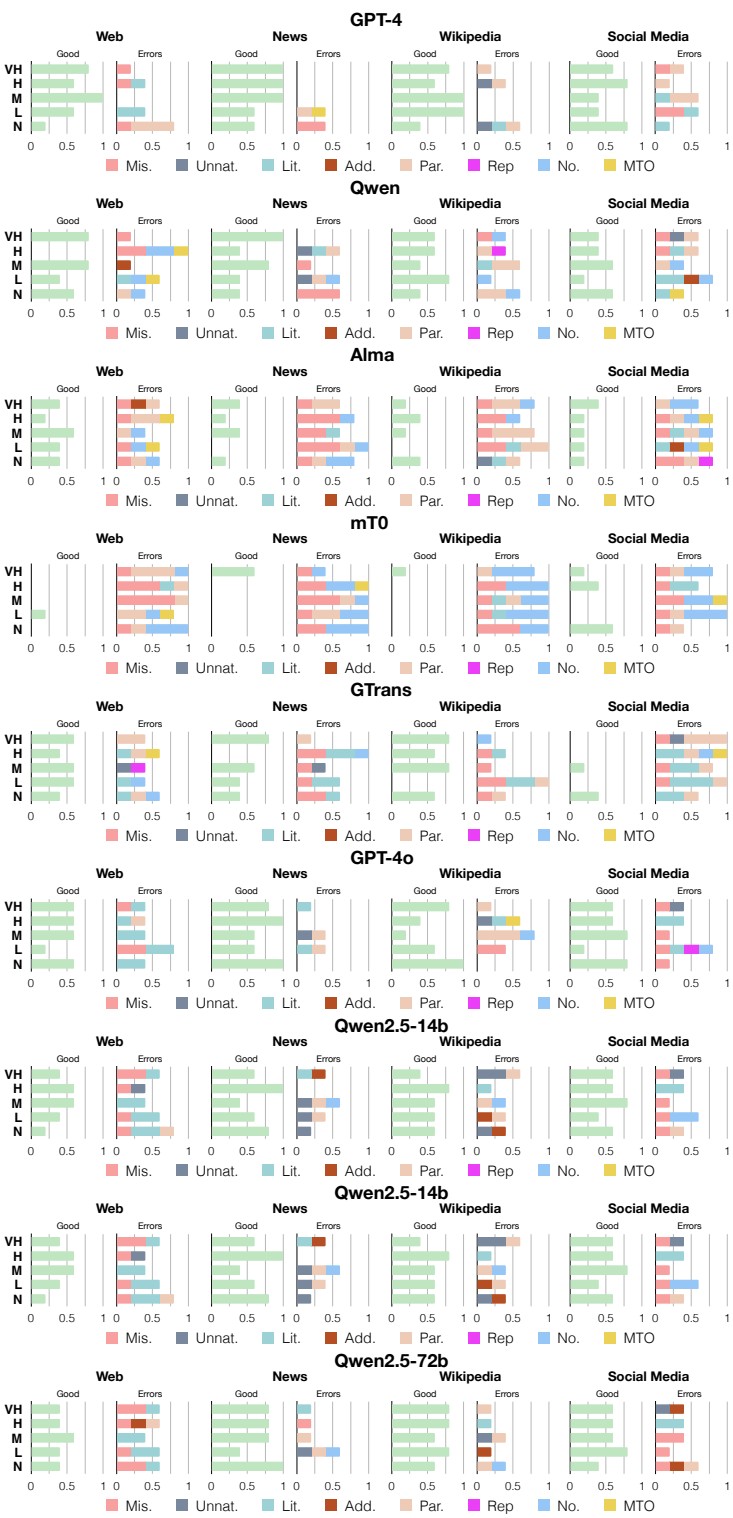

Figure 9: Frequency-centric view of translation results in each domain. There are two subfigures in each dataset. Left subfigure: mean ratio of good translations across all systems. Right subfigure: mean ratio of error categories across all systems.

# E  Further Analysis on Evaluation Metrics

## E.1  Kendall's $\tau$ between Evaluation Metrics and Human Annotations

Table 9 shows Kendall's $\tau$ between evaluation metrics and human annotations. Similar to Table 3, Kendall's $\tau$ remains weak across categories, indicating low agreement between ranking based on metric scores and human ratings.

| Scope | Category | Size | Reference-Based Metrics | | | | | Reference-Free Metrics | | |
|---|---|---|---|---|---|---|---|---|---|---|
| | | | BLEU | BERTScore | COMET | MetricX-23-XXL | MetricX-24-XXL | COMETKIWI | MetricX-QE-23-XXL | MetricX-QE-24-XXL |
| Full | Mistranslation | 128 | 0.064 | 0.099 | 0.141 | 0.081 | 0.129 | 0.192 | 0.049 | 0.097 |
| | Literal | 90 | 0.153 | 0.212 | 0.314 | 0.248 | 0.210 | 0.091 | 0.158 | 0.040 |
| | Partial | 105 | 0.176 | 0.175 | 0.238 | 0.098 | 0.158 | 0.097 | 0.240 | 0.247 |
| | No Translation | 74 | 0.182 | 0.271 | 0.345 | 0.204 | 0.232 | 0.072 | 0.156 | 0.273 |
| Idiom | Mistranslation | 128 | -0.011 | 0.030 | 0.196 | 0.210 | 0.181 | 0.184 | 0.120 | 0.123 |
| | Literal | 90 | -0.005 | 0.078 | 0.143 | 0.216 | 0.082 | 0.016 | -0.052 | -0.102 |
| | Partial | 105 | 0.122 | 0.104 | 0.072 | 0.034 | -0.107 | -0.043 | -0.171 | -0.195 |
| | No Translation | 74 | * | * | -0.016 | -0.019 | -0.269 | 0.031 | -0.273 | -0.252 |
| Full | All errors | 452 | 0.199 | 0.269 | 0.323 | 0.199 | 0.211 | 0.173 | 0.155 | 0.162 |
| | All categories | 900 | 0.208 | 0.274 | 0.285 | 0.264 | 0.246 | 0.140 | 0.137 | 0.147 |
| Idiom | All errors | 452 | 0.178 | 0.218 | 0.259 | 0.269 | 0.006 | 0.012 | -0.078 | -0.140 |
| | All categories | 900 | 0.309 | 0.311 | 0.373 | 0.414 | 0.093 | 0.016 | -0.054 | -0.090 |

Table 9: Kendall's $\tau$ between evaluation metrics and human annotations. Evaluation is measured on both *Full* translation and *Idiom* translation. Good is omitted since it does not have severity scores. More Than One is omitted here as all its instances are rated as the highest severity. Unnatural, Addition and Repetition are omitted due to small sample size. *: for No Translation on *Idiom*, BLEU and BERTScore outputs 0 and thus omitted. Existing metrics fail to produce measures that are correlated strongly with human annotations.

## E.2  Prompts

Figure 10 shows prompt for extracting idiom translations. Figure 11 shows prompt for editing idiom translations. Figure 12 shows prompt for identifying idiom translations.

# F  Implementation Details for Good and Bad Idiom Translation Detection

## F.1  Prompting Methods

We use the latest GPT-4o-20241120 (referred to as GPT-4o) and Qwen2.5 Instruct models for prompting. Generation is done via greedy decoding. Figure 13 displays the zero-shot prompt, and Figure 14 displays the zero-shot prompt with chain-of-thought.

## F.2  Fine-tuning Methods

We fine-tune Qwen2.5 Base models using LoRA (Hu et al., 2022), and use greedy decoding during inference. For MetricX-24, we add a classification head to the decoder and update both the LoRA parameters and the classification head during training. We sweep learning rates in {5e-5, 1e-4, 3e-4, 5e-4} for 3 epochs and select the best checkpoint on the validation set for final testing.

**Input Format for Fine-tuning Qwen2.5:**

```
Evaluate whether the Chinese idiom is correctly translated in the following text:

- Chinese Idiom: {idiom}
- Chinese Text: {zh_sentence}
- English Translation: {en_sentence}
```

**Output Formats of Different Strategies for Fine-tuning Qwen2.5:**

1. **Short Answer**
   **Template:**
   ```
   yes  /  no
   ```
   *Description:* The model outputs a single token indicating correctness ("yes") or incorrectness ("no").

2. **Long Answer**
   **Template:**
   ```
   the idiom {idiom} got translated {correctly  /  incorrectly}.
   ```
   *Description:* The model outputs one short sentence describing whether the idiom translation is correct or incorrect, but does not include any reasoning.

3. **Short Thought Process + Final Answer**
   **Template:**
   ```
   The translation of the idiom is "{idiom_translation}".  Given the
   context, I think this is a {category}, so my final answer is:  the
   idiom got translated {decision}.
   ```
   *Description:* The model provides a brief rationale (e.g., mentioning the idiom translation) and then states whether it is correct or incorrect.

4. **Long Thought Process + Final Answer**
   **Template:**
   ```
   Analyzing the translation of "{idiom}" as "{idiom_translation}". When
   examining this translation: {definition based on category} Therefore,
   I conclude this is (not) a good translation. The idiom "{idiom}" has
   been translated {decision}.
   ```
   *Description:* The model generates a more elaborate thought, referencing predefined definitions for the relevant error category (or "Good Translation"). It concludes by stating whether the translation is correct or incorrect.

**Prompt for Extracting Idiom Translations**

```
## Task
Analyze the provided Chinese idiom, Chinese sentence containing the idiom
↪   and its English translation to extract corresponding idiom
↪   translation.

## Input to Analyze
* Chinese idiom: [PLACEHOLDER]
* Chinese sentence: [PLACEHOLDER]
* English translation: [PLACEHOLDER]

Please note:
1. Ensure the extracted idiom translation is short and concise, and do
↪   not include irrelevant translation.
2. If the given idiom appears multiple times in the Chinese sentence,
↪   only analyze the first occurrence.
3. If no corresponding translation is found, output an empty string.

Read the input carefully. Write down a brief thought process first, with
↪   the 3 notes in mind. Then extract the translation of the idiom.
Respond in a JSON format {"Idiom translation": }. The key must be "Idiom
↪   translation", and the value must be the corresponding idiom
↪   translation in the English sentence.
```

Figure 10: Prompt for Extracting Idiom Translations

---

**Prompt for Perturbing Idiom Translations**

```
## Task
Analyze the provided Chinese idiom, idiom meaning, Chinese sentence
↪   containing the idiom and its English translation.
The goal is to edit the given idiom translation to comply with the given
↪   category.

## Categories
There are 7 categories to consider:
* Mistranslation: idiom translation is incorrect due to wrong choices of
↪   words or phrases, and it affects our understanding of the translated
↪   sentence.
* Partial Translation: idiom is translated partially. Part of the idiom
↪   meaning is missing or the extremity is inaccurate.
* Repetition: translation of the idiom contains repeated words or phrases,
↪   or their synonyms.
* Unnatural: the translation is ok but not perfect enough due to improper
↪   choices of words or grammar errors. In other words, the translation
↪   can be improved by more appropriate choices of words.
* Literal Translation: idiom is translated literally and translation is
↪   not coherent with context.
* Addition: In addition to good translation of idiom, the translation
↪   also contains non-present content in the source.
* No Translation: there is no translation of the idiom in the output.

## Input to Edit
* Chinese idiom: [PLACEHOLDER]
* Idiom meaning: [PLACEHOLDER]
* Chinese sentence: [PLACEHOLDER]
* English translation: [PLACEHOLDER]
* Idiom translation: [PLACEHOLDER]

Please note:
1. Given category only applies to the idiom translation itself.
2. An idiom can have multiple meanings. Find the most appropriate meaning
↪   under the given context.

Read the input carefully. YOU NEED TO MAKE THE IDIOM TRANSLATION A
↪   [PLACEHOLDER].
First, modify the idiom translation to comply with the given category.
Then, rewrite the given translation by replacing the original idiom
↪   translation with the new translation.

Do not include your thought process.

Output the corresponding edited translation only. Do not include anything
↪   else in your output.
```

Figure 11: Prompt for Perturbing Idiom Translations

Prompt for Editting Idiom Translations

```
## Task
Analyze the provided Chinese idiom, idiom meaning, Chinese sentence
↪   containing the idiom and its English translation to identify idiom
↪   translation category.

## Categories
There are 8 categories to consider:
* Good Translation: the idiom is translated perfectly.
* Mistranslation: idiom translation is incorrect due to wrong choices of
↪   words or phrases, and it affects our understanding of the translated
↪   sentence.
* Partial Translation: idiom is translated partially. Part of the idiom
↪   meaning is missing or the extremity is inaccurate.
* Repetition: translation of the idiom contains repeated words or phrases,
↪   or their synonyms.
* Unnatural: the translation is ok but not perfect enough due to improper
↪   choices of words or grammar errors. In other words, the translation
↪   can be improved by more appropriate choices of words.
* Literal Translation: idiom is translated literally and translation is
↪   not coherent with context.
* Addition: In addition to good translation of idiom, the translation
↪   also contains non-present content in the source.
* No Translation: there is no translation of the idiom in the output.

## Input to Analyze
* Chinese idiom: [PLACEHOLDER]
* Idiom meaning: [PLACEHOLDER]
* Chinese sentence: [PLACEHOLDER]
* English translation: [PLACEHOLDER]
* Idiom translation: [PLACEHOLDER]

Please note:
1. Focus on analyzing the translation of the idiom under the context of
↪   the sentence.
2. Compare the idiom meaning and provided idiom translation to make the
↪   judgement. Note an idiom can have multiple meanings.
3. If the given idiom appears multiple times in the Chinese sentence,
↪   only analyze the first occurrence.
4. If the idiom translation only captures the essence of the meaning, it
↪   is a "Partial Translation".

Read the input carefully. Write down a brief thought process first, with
↪   the 4 notes in mind. Then identify the translation category of the
↪   idiom.
Respond in a JSON format: the keys must be "Category", and the values
↪   must be the corresponding category of the idiom translation.
```

Figure 12: Prompt for editting idiom translations

```
Zero-shot Prompt for Idiom Translation Correctness Detection

Evaluate whether the Chinese idiom is correctly translated in the
↪  following text:

- Chinese Idiom: {idiom}
- Chinese Text: {zh_sentence}
- English Translation: {en_sentence}

Note: only output Yes or No in your response. Do not include anything
↪  else.
```

Figure 13: Zero-shot Prompt for Idiom Translation Correctness Detection

```
Zero-shot with CoT Prompt for Idiom Translation Correctness Detection

Evaluate whether the Chinese idiom is correctly translated in the
↪  following text:

- Chinese Idiom: {idiom}
- Chinese Text: {zh_sentence}
- English Translation: {en_sentence}

Note: Analyze this step by step with the following output format:
- Thought Process: {{your analysis of the idiom's meaning and translation
↪  accuracy}}
- Final Answer: {{correct translation / wrong translation}}
```

Figure 14: Zero-shot with CoT Prompt for Idiom Translation Correctness Detection

