# OpenReview forum: "Evaluating LLMs on Chinese Idiom Translation"
_colmweb.org/COLM/2025/Conference — COLM 2025_

### Official Review · Reviewer_u3jM · 2025-05-10

**Rating:** 9
**Confidence:** 4
**Ethics Flag:** 1

**Summary:**

This paper introduces an evaluation framework for identifying errors in Chinese idiom translation. They apply it to compare the performance of nine contemporary LLM-based systems, to translate idioms in multiple domains. The results show that these systems frequently mistranslate idioms, while existing evaluation metrics poorly assess idiom quality. The authors describe and extensive evaluation and propose alternative metrics to alleviate such an issue.

The paper is very clear and well written. The presented analysis is quite complete and well motivated. They tackle the problem (idioms translation) from the evaluation perspective, performing a comprehensive comparison between LLM-based SoA systems, proposing a new taxonomy and evaluation framework. Interesting conclusions are drawn, most of them are as expected but now they are supported by evidence. One of them is surprising, though, which is that the performance increases as the model scales up, except at 72B. It would have been interesting to hear a hypothesis about this behavior in the paper.

The proposed evaluation metric correlates better with human opinion than current ones. My impression, not being an expert in MT myself, is that the proposed framework is general enough to be applied to other languages.

Despite the paper is very clear, there are a few minor issues to be fixed (see below)

**Questions To Authors:**

Only one:
* why do you think that the performance increases as the model scales up except at 72B? (no need to prove the answer, but just to formulate an hypothesis from your expert view)

**Reasons To Accept:**

* The paper is very clear and well written.
* The presented analysis is quite complete and well motivated.
* Interesting conclusions, which highlight the difficulties faced by LLMs in tasks as difficult as capturing idioms.
* The proposed evaluation metric correlates better with human opinion than current ones.

**Reasons To Reject:**

I see NO reasons to reject. I write here some minor issues to be fixed:

- In the introduction, there is a missing closing parenthesis in "... (see Table 1"
- Fonts in some figures (e.g., figure 2, and many other in the annex) are too small and not easily readable in printed version
- IDIOMEVAL is sometimes introduced as a taxonomy (e.g., the abstract) and other times as a framework (e.g., the introduction). Authors need to be consistent with this. My understanding it that it is a framework that, among other things, defines a taxonomy.

---

> ### Author Response · Authors · 2025-06-03
> **Author Response**
>
> Thank you for your valuable feedback and high assessment of our work. We appreciate your helpful suggestions.
>
> > In the introduction, there is a missing closing parenthesis in "... (see Table 1"
>
> Thank you for catching this. We will fix this in the final version.
>
> > Fonts in some figures (e.g., figure 2, and many other in the annex) are too small and not easily readable in printed version
>
> We will increase the font size in these figures to improve the readability.
>
> > IDIOMEVAL is sometimes introduced as a taxonomy (e.g., the abstract) and other times as a framework (e.g., the introduction). Authors need to be consistent with this. My understanding it that it is a framework that, among other things, defines a taxonomy.
>
> You are right. IDIOMEVAL is best described as a framework that includes a taxonomy for evaluating idiom translation. We will revise the wording throughout the paper to ensure consistent use of the terminology.
>
> > Why do you think that the performance increases as the model scales up except at 72B? (no need to prove the answer, but just to formulate an hypothesis from your expert view)
>
> Thank you for raising this point. We further investigate the cause of this case. In terms of training loss (cross-entropy), we found that when fine-tuned on *thoughts*, the 72B model plateaus around 0.2 to 0.3, while smaller models (32B, 14B, 7B) converge more smoothly to a lower loss (~0.02). In contrast, training the 72B model on *answers only*, especially short answers, results in lower loss and better performance.
>
> In terms of model behavior, we find that the 72B model always outputs the *short answer format (“yes” / “no”)* even when fine-tuned on longer formats such as long answers or thought-based responses. This differs from all the smaller size models (1.5B to 32B), which follow the training output format. While we do not fully understand the root cause of this behavior, it may explain why training the 72B model on the *short answer* produces the best performance.
>
> We will include this explanation in the final version.
>
> ---
> Please let us know if you have any remaining questions or concerns.

---

> > ### Comment · Reviewer_u3jM · 2025-06-03
> >
> > I thank the authors for their response. I keep my scores and (positive) overall assessment

---

### Official Review · Reviewer_j3Fh · 2025-05-12

**Rating:** 6
**Confidence:** 4
**Ethics Flag:** 1

**Summary:**

In this paper, an evaluation framework (i.e., IDIOMEVAL) is introduced as a new taxonomy of Chinese idiom translation errors. Using this framework, a high-quality dataset of 900 human-annotated translations based on four domains and nine LLMs is collected. The result of experiments using this dataset indicated that the translation qualities of nine modern systems and the evaluation qualities of existing metrics and prompting approaches are insufficient. Moreover, this paper proposed new evaluation models by instruction-tuning LLMs for improvement of evaluation of Chinese Idiom translation, and showed that the proposed models which are instruction-tuned LLMs outperform existing metrics and prompting approaches.

**Reasons To Accept:**

The advantages of the paper is as follows:

1. This paper indicated that the modern translation systems are insufficient in Chinese idiom translation of some domains using high-quality dataset based on human-annotated translations.

2. This paper indicated that the evaluation quality of existing metrics and prompting approaches is low based on the coefficient correlation with human judgements (i.e., under 0.48) in Chinese idiom translation.

3. This paper proposed the instruction-tuned LLMs using dataset by human-annotated translations to detect good and bad idiom translation as a binary classification problem. Moreover, though the experimental results, it is indicated that Macro F1 of the proposed models (i.e., 0.68) are higher than metrics and prompting approaches.

**Reasons To Reject:**

The effectiveness of the proposed approach is not clear for the following points;

1. In the proposed models, native Chinese speakers fluent in English as annotators are required to obtain the dataset for instruction-tuning. Therefore, the annotators and cost are serious problems.

2. In detection of idiom translation errors of Table 6, it is not clear whether there are statistical significances or not in F1-Macro between models.

3. The reference-based or reference-free metrics are formulated as a binary classification task using ROC curve. However, it may be a little unfair because the metrics except BLEU based on n-gram matching is not learned to perform a binary classification task.

4. This paper needs to evaluate idiom translation of other languages, not only Chinese.

---

> ### Author Response · Authors · 2025-06-03
> **Author Response**
>
> Thank you for your valuable feedback on our work. We appreciate the opportunity to address the points you raised.
>
> > In the proposed models, native Chinese speakers fluent in English as annotators are required to obtain the dataset for instruction-tuning. Therefore, the annotators and cost are serious problems.
>
> Yes, annotating the translations is costly, at around $4K USD, which is expected for high-quality, fine-grained annotations. To avoid repeated annotation efforts in future work, we will release our datasets and models publicly so that others can directly build on our work to improve idiom translation performance and develop idiom-aware evaluation metrics. For example, our fine-tuned model can be used for self-training or back-translation to facilitate transfer to other languages.
>
> > In detection of idiom translation errors of Table 6, it is not clear whether there are statistical significance or not in F1-Macro between models.
>
> Thank you for the suggestion. We conduct one-sided paired bootstrap significance tests (10,000 resamples) to compare each metric’s macro F₁ score against that of the best-performing model, fine-tuned Qwen2.5-72B. We report the 95% confidence interval of the macro F₁ and p-value. The p-value reflects the proportion of bootstrap samples in which Qwen2.5-72B achieved a higher macro F₁ than the compared model. An asterisk (*) indicates statistical significance at the 0.05 level (p < 0.05). To save space, we show only the top-performing metrics from each method category. This significance analysis will be added to Table 6 in the final version.
>
> | Model                         | Params   |   Macro F₁ | 95 % CI          | p-value   |
> |:------------------------------|:---------|-----------:|:-----------------|:----------|
> | **Thresholded Metrics (ROC)** |
> | COMET                      | 580 M    |     0.5762 | [0.5276, 0.6246] | 0.0045*   |
> | MetricX-23                | 13 B     |     0.6074 | [0.5582, 0.6548] | 0.0410*   |
> | MetricX-24                | 13 B     |     0.5983 | [0.5487, 0.6454] | 0.0235*   |
> | **Prompting LLMs** |
> | Gpt-4o (zero-shot)                     | —        |     0.6271 | [0.5764, 0.6744] | 0.0989    |
> | Qwen2.5 (zero-shot CoT)               | 14 B     |     0.6037 | [0.5429, 0.6426] | 0.0122*   |
> |GPT-4o (zero-shot CoT)                   | —        |     0.5960 | [0.5433, 0.6459] | 0.0163*   |
> | **Fine-tuned Models** |
> | Qwen2.5         | 14 B     |     0.6543 | [0.6041, 0.7029] | 0.3579    |
> | Qwen2.5              | 72 B     |     **0.6653**  | [0.6176, 0.7112] | —         |
>
>
> > The reference-based or reference-free metrics are formulated as a binary classification task using ROC curve. However, it may be a little unfair because the metrics except BLEU based on n-gram matching is not learned to perform a binary classification task.
>
> We agree that traditional translation metrics such as COMET and MetricX are not trained for binary classification. That’s exactly why we apply the ROC curve to determine a threshold on the training set, allowing us to convert their raw scores into binary decisions for fair comparison with prompting and fine-tuned methods on the test set.
>
> In addition, Table 3 displays the correlation between their raw metric scores and human judgements. Together with the binary classification results, these provide a comprehensive picture of how traditional metrics perform in detecting good and bad idiom translations.
>
> > This paper needs to evaluate idiom translation of other languages, not only Chinese.
>
> We agree that applying our taxonomy to other languages is exciting. But considering the substantial annotation effort ($4K USD) and author’s expertise in Chinese, we decide to focus on Chinese idioms, which have the uniqueness of proverbs, allegorical sayings, and aphorisms originated from ancient literature and is overlooked in prior work [Line 26-37]. Given budget constraints, we leave the extension to other languages to future work.
>
>
> ---
> Please let us know if you have any remaining questions or concerns.

---

> > ### Author Response · Authors · 2025-06-08
> >
> > Hello Reviewer j3Fh, thank you again for your thoughtful feedback. As the discussion period comes to an end, we just want to check if you have any remaining questions. If our responses have addressed your concerns, we would greatly appreciate it if you could consider revisiting your score.

---

> > > ### Comment · Reviewer_j3Fh · 2025-06-10
> > >
> > > Thank you for your response. The weakness I mentioned has been generally resolved. Therefore, I keep my positive scores.

---

### Official Review · Reviewer_9war · 2025-05-12

**Rating:** 6
**Confidence:** 4
**Ethics Flag:** 1

**Summary:**

This paper addresses the evaluation of machine translation systems with a focus on Chinese idioms, which are rich in figurative meaning and deeply embedded cultural context. The authors introduce IDIOMEVAL, a comprehensive error taxonomy designed to assess the accuracy of idiomatic translation.
Since existing metrics have a low correlation with human judgment, they developed an improved model that detects idiom translation errors.

**Questions To Authors:**

+ As mentioned above, the “Results” section would benefit from a more focused analysis. Consider narrowing the scope of the key arguments to allow for deeper discussion.

+ One suggestion is to move the CoT experiment details to the appendix, thereby freeing up space to expand on the qualitative evaluation—perhaps by including more illustrative examples of idiom translation errors and model behavior.

**Reasons To Accept:**

+ This paper presents the potential of LLMs in the nuanced evaluation of idiomatic translations, while also revealing that LLM systems make critical errors.

+ By focusing on Chinese idiom, which is a relatively underexplored are, the authors contribute a framework that systematically categorizes translation errors in a way that reflects the linguistic and cultural complexity of Chinese.

+ This study is comprehensive in scope, encompassing data collection, error taxonomy design, and evaluation methodology, and it appears to promise for adaptation to other languages.

**Reasons To Reject:**

+ While the analysis involving major LLMs and CoT prompting is valuable, the experimental section feels limited in depth due to space constraints. There is a lack of variation and exploration across different conditions.

+ For instance, in Figure 4, the performance drop at the 72B scale for both “short thought with answer” and “long thought with answer” prompts is noted, yet the paper does not sufficiently explore or explain the underlying causes of this decline.

---

> ### Author Response · Authors · 2025-06-03
> **Author Response**
>
> Thank you for your valuable feedback on our work. We appreciate the opportunity to address the points you raised.
>
> > In Figure 4, the performance drop at the 72B scale for both “short thought with answer” and “long thought with answer” prompts is noted, yet the paper does not sufficiently explore or explain the underlying causes of this decline.
>
> Thank you for raising this point. We further investigate the cause of this case. In terms of training loss (cross-entropy), we found that when fine-tuned on *thoughts*, the 72B model plateaus around 0.2 to 0.3, while smaller models (32B, 14B, 7B) converge more smoothly to a lower loss (~0.02). In contrast, training the 72B model on *answers only*, especially short answers, results in lower loss and better performance.
>
> In terms of model behavior, we find that the 72B model always outputs the *short answer format (“yes” / “no”)* even when fine-tuned on longer formats such as long answers or thought-based responses. This differs from all the smaller size models (1.5B to 32B), which follow the training output format. While we do not fully understand the root cause of this behavior, it may explain why training the 72B model on the *short answer* produces the best performance.
>
> We will include this explanation in the final version.
>
> > As mentioned above, the “Results” section would benefit from a more focused analysis. Consider narrowing the scope of the key arguments to allow for deeper discussion. One suggestion is to move the CoT experiment details to the appendix, thereby freeing up space to expand on the qualitative evaluation—perhaps by including more illustrative examples of idiom translation errors and model behavior.
>
> We agree that including more detailed analysis in the main paper would be beneficial, though we were limited by space constraints. The current version includes example translation errors in Appendix A, and we will expand this with additional examples from each model. If accepted, we will also move some detailed analysis in Appendix D, such as common error types per model and error distribution by frequency, into the main content using the extra page.
>
> ---
> Please let us know if you have any remaining questions or concerns.

---

> > ### Comment · Reviewer_9war · 2025-06-05
> > **Thank you for the response.**
> >
> > Thank you for your detailed response to my review. I appreciate the clarifications provided and trust that your explanations will be added into the revised version of the paper. Accordingly, I maintain my positive evaluation scores.

---

### Official Review · Reviewer_vzbR · 2025-05-13

**Rating:** 6
**Confidence:** 4
**Ethics Flag:** 1

**Summary:**

The paper focuses on translation of Chinese idioms. The authors propose an error taxonomy, create an annotated dataset and evaluate how well existing MT metrics capture errors in the translation of idioms. Finally, experiments are carried out with the goal of improving the idiom translation evaluation.

**Questions To Authors:**

The taxonomy is documented in different places throughout the paper and the number of categories varies. At first you mention _7 different failure modes_ and Table 1 indeed contains 7 error categories. However, then you mention 9 categories and 13 subcategories. Which version is correct?

I would suggest using "minor", "major" and "critical" for severity. For annotator confidence, something like "low", "medium", "high" might work better as well.

**Reasons To Accept:**

* The annotation interface, the dataset and evaluation methodology can potentially be useful to other researchers.
* The paper is relatively well written and easy to follow.
* The scope of the evaluation is relatively large, including various SotA translation systems/LLMs and evaluation metrics.

**Reasons To Reject:**

* There is not much novelty in the work; the paper mostly just introduces a dataset and the results of an evaluation.
* Limited language coverage, focusing only on Chinese idioms.
* Various inaccuracies and confusing parts in the text (see questions to authors).
* I have some doubts about the improved methods for evaluation; based on the training data description it looks like the same idioms were present in both training and evaluation, which casts doubt on the positive result.

---

> ### Author Response · Authors · 2025-06-03
> **Author Response**
>
> Thank you for your valuable feedback on our work. We appreciate the opportunity to address the points you raised.
>
> > There is not much novelty in the work; the paper mostly just introduces a dataset and the results of an evaluation.
>
> While our paper follows a common format for evaluation work, we believe the novelty of our work lies in the focus and contributions. We introduce a new taxonomy for Chinese idiom translation, provide a high-quality annotated dataset across four domains and nine systems, and conduct the first systematic analysis of both translation models and metrics on this task. In addition, we go beyond the dataset and analysis by training models that outperform existing metrics in detecting idiom translation errors.
>
> > Limited language coverage, focusing only on Chinese idioms.
>
> We agree that applying our taxonomy to other languages is exciting. But considering the substantial annotation effort ($4K USD), authors’ expertise in Chinese, and the unique nature of CHinese idioms—which often include proverbs, allegorical sayings, and aphorisms originating from ancient literature and differ from those from prior work studying English, Japanese and European languages [Line 26-37]—we decide to focus on Chinese idioms. Given budget constraints, we leave the extension to other languages to future work.
>
> > I have some doubts about the improved methods for evaluation; based on the training data description it looks like the same idioms were present in both training and evaluation, which casts doubt on the positive result.
>
> This is a great point. In fact, only one idiom out of 100 idioms in the test set appears in the training set. This minimal overlap is due to the data split: test set translations are based on 2024 data, while the train/val sets are based on 2023 data [see Line 92–95], leading to different sampled idioms. The train/val split is also done at the idiom level to prevent overlap. We will make this clearer in the final version.
>
> To show the impact of the one overlapping idiom, we present a side-by-side comparison of the original Macro F₁  in Table 6 and updated Macro F₁ after removing the one overlapping idiom from the test set below. To save space, we report only the top-performing metrics from each method category. The differences are very minor across all metrics, and the fine-tuned models show even smaller variation than some non-fine-tuned baselines.
>
> | Model | Params | Macro F₁ (Orig.) | Macro F₁ (New) | Δ (New−Orig.) |
> |-------|--------|------------------|----------------|---------------|
> | **Thresholded Metrics (ROC)** |  |  |  |  |
> | COMET | 580 M | 0.5762 | 0.5777 | +0.0015 |
> | MetricX-23 | 13 B | 0.6074 | 0.6076 | +0.0002 |
> | MetricX-24 | 13 B | 0.5983 | 0.5963 | −0.0020 |
> | **Prompting LLMs** |  |  |  |  |
> | GPT-4o (zero-shot) | — | 0.6271 | 0.6260 | −0.0011 |
> | Qwen2.5 (zero-shot CoT) | 14 B | 0.6037 | 0.6033 | −0.0004 |
> | GPT-4o (zero-shot CoT) | — | 0.5960 | 0.5941 | −0.0019 |
> | **Fine-tuned Models** |  |  |  |  |
> | Qwen2.5 | 14 B | 0.6543 | 0.6534 | −0.0009 |
> | Qwen2.5 | 72 B | **0.6653** | **0.6646** | −0.0007 |
>
>
> > The taxonomy is documented in different places throughout the paper and the number of categories varies. At first you mention 7 different failure modes and Table 1 indeed contains 7 error categories. However, then you mention 9 categories and 13 subcategories. Which version is correct?
>
> Both are correct, let us explain. Table 1 displays 8 categories: 1 for *No Error* and 7 for different error types. As noted in the Table 1 caption, we also have a  *More than One* category for translations containing multiple errors, bringing the total to 9 categories. Three of these categories have subcategories:  *No Error* has 2, *Partial Translation* has 3, and *No Translation* has 2. The remaining 6 categories are stand-alone categories. Counting both the subcategories and stand-alone categories, we have 13 subcategories in total. Please see Appendix A for the full list. We will make this clearer in the final version.
>
> > I would suggest using "minor", "major" and "critical" for severity. For annotator confidence, something like "low", "medium", "high" might work better as well.
>
> Thank you for your suggestion. We will use these terms to improve clarity in the final version.
>
> ---
> Please let us know if you have any remaining questions or concerns.

---

> > ### Author Response · Authors · 2025-06-08
> >
> > Hello Reviewer vzbR, thank you again for your thoughtful feedback. As the discussion period comes to an end, we just want to check if you have any remaining questions. If our responses have addressed your concerns, we would greatly appreciate it if you could consider updating your score.

---

> > > ### Comment · Reviewer_vzbR · 2025-06-10
> > >
> > > Thank you for the thorough response. Because it meaningfully addresses some of the core concerns I had, I'm increasing the overall rating.

---

### Decision · Program_Chairs · 2025-07-08

**Decision:**

Accept

**Comment:**

This paper introduces an evaluation framework for Chinese idiom translation that could potentially be extended to other languages (9war, u3jM). It includes a supporting dataset (with 900 translation pairs annotated for evaluation) and taxonomy, evaluates a range of modern MT/language models, and shows that existing metrics correlate poorly with human judgments. Finally, it develops models that achieve better Chinese idiom translation results.

As for reasons to accept: the paper is clear and well written (j3FH, u3JM), and the dataset and evaluation methodology for Chinese idiom translation (which is relatively underexplored, 9war), including taxonomy, are quite comprehensive could be useful to other researchers (vzbR) and could be extended to other languages (9war, u3JM). The paper highlights the limitations of existing translation models and metrics with interesting results (j3FH, u3JM) and its (revealing) analysis is seen as a strong point (9war, u3JM).

As for reasons to reject, two reviewers would have liked the approach to be applied to languages beyond Chinese (vzbR, j3FH) (even though the authors reasonably make it clear that a lot of expertise is needed for idioms and annotation costs are high, so it seems reasonable to focus on just one language pair here, if compensated by e.g. a strong analysis). Only one reviewer finds the paper limited in novelty (vzbR), and one finds the experimental section somewhat shallow (9war). One reviewer (out of four) does not find any reasons to reject the paper. In author discussion it became clear that one idiom was both in the train and test set, and the authors showed that this had minimal impact on the results.

Even though idiom translation, especially using automatic systems, is relatively underexplored, I find that the paper does not situate itself very well in existing literature (its related work section is shallow and it hardly interacts with that work), and its main novelties are its mined dataset and (detailed) focus on Chinese and reporting idiom translation results and analyses of modern systems like GPT-4o, including with the metrics analysis. There is still a lot of value in that, and the reviewers found ample reasons to (weakly) accept it despite this limitation, but the paper would have been stronger had it built more on previous work. Overall, three reviewers suggest this is marginally above the acceptance threshold, and based on the above I believe it is just that.